# Screening of *Codonopsis radix* Polysaccharides with Different Molecular Weights and Evaluation of Their Immunomodulatory Activity In Vitro and In Vivo

**DOI:** 10.3390/molecules27175454

**Published:** 2022-08-25

**Authors:** Guangxin Li, Yanhong Ju, Yuwei Wen, Min Zuo, Chao Wang, Xiaomei Zhang, Xiaoxiang Hou, Guiqing Yang, Jianping Gao

**Affiliations:** 1College of Agriculture, Shanxi Agricultural University, Taiyuan 030031, China; 2Modern Research Center for Traditional Chinese Medicine, Shanxi University, Taiyuan 030006, China; 3School of Pharmacy, Shanxi Medical University, Taiyuan 030001, China

**Keywords:** *Codonopsis radix*, polysaccharide, immune activity

## Abstract

Polysaccharide is one of the main components of *Codonopsis radix* (*CR*) and has good immune activity. However, the immune activity of *CR* polysaccharides with different molecular weights has not been systematically screened. In this study, the polysaccharides of *CR* from Pingshun of Shanxi Province (PSDSs) were first divided into two groups using ultrafiltration: 3.3 kDa (PSDSs-1) and more than 2000 kDa (PSDSs-2). The immunomodulatory effects of PSDSs with different molecular weights were evaluated in vitro and in vivo. In vitro experimental results showed that compared with Lipopolysaccharide-induced macrophages, PSDSs-1 increased TNF-α and IL-6 levels and decreased IL-10. Meanwhile, PSDSs-2 showed the opposite effect, indicating the difference in pro- and anti-inflammatory activities of PSDSs with different molecular weights. The immunosuppressive model of cyclophosphamide proved that PSDSs have immune-promoting function, with PSDSs-1 exhibiting a better effect than PSDSs-2. In vitro and in vivo experiments illustrated the complexity of PSDS immunomodulation. Further research on the functions of PSDs with different molecular weights is needed to lay a foundation for their classification and application.

## 1. Introduction

*Codonopsis radix* (*CR*) is the dried root of *Codonopsis pilosula* (Franch.) Nannf., *C. pilosula* Nannf. var. modesta (Nannf.) L.T. Shen and *Codonopsis tangshen* Oliv., and has the ability to strengthen the spleen and lung and nourish the blood and body fluids; it is used to treat various syndromes, such as deficiency of spleen–lung Qi, appetite loss and listlessness, cough, asthma and deficiency of Qi and blood [1]. *Ludangshen* is the dried root of *C. pilosula* (Franch.) Nannf., which has a long cylindrical shape with a grey-yellow, yellow-brown to grey-brown surface, a ‘lion panhead’ and other features. This genuine medicinal material is mainly produced in the Shanxi Province, including Huguan, Pingshun of Changzhi and Lingchuan of Jincheng and its surrounding areas [2].

Polysaccharide is one of the main components of *CR*, possessing immunity enhancing, anti-aging and blood sugar-lowering effects [3,4]. Although many studies have been conducted on the immunoregulatory properties of *CR* polysaccharides at home and abroad, these studies mainly focused on *CR* total polysaccharides. Systematic studies have been carried out on the extraction, structural analysis and immunoregulatory properties of *CR* total polysaccharides. Additionally, the relationship between structural modification and immune regulation has been investigated. Many reports have focused on the structure of *CR* total polysaccharides, especially their molecular weight (Mw) and monosaccharide composition; however, only a few systematic studies have been performed to gather data on other aspects, such as glycosidic bonds and connecting sequences. In terms of pharmacological activities, *CR* total polysaccharides have strong immunomodulatory effects, which can improve the ability of immune organs, immune cells and immune molecules to perform their immune activities [5]. Fu et al. [6] and Li et al. [7] reported that *CR* polysaccharides could significantly improve the performance of the thymus, spleen, liver and intestinal lymph node index of cyclophosphamide (CTX)-induced immunosuppressive mice and haemorrhagic rabbits. Meng et al. [8] obtained CP1-2-1 and CP3-1-1 polysaccharides using a gel column and studied their structural characteristics and anti-inflammatory activities. Wang [9] screened 1698 Da *CR* polysaccharide using ultrafiltration membrane separation and studied its structure and in vitro immune activity. However, the isolation and preparation of *CR* polysaccharides with different Mws and the systematic screening of their immune activity are seldom reported. In the present work, ultrafiltration membrane separation was used to purify *CR* polysaccharides with different Mws. Their immune activity was studied in vitro and in vivo to clarify the pharmacodynamics of *CR* polysaccharides.

High-performance liquid chromatography-refractive index detection (HPLC-RID) was used to characterise the Mw distribution of *CR* polysaccharides. Through the degradation of polysaccharides combined with pre-column derivatization, HPLC-ELSD and HPLC-UV were employed to analyse the monosaccharide composition of *CR* polysaccharides through acid degradation and pre-column derivatisation. *CR* polysaccharides with different Mws were prepared by ultrafiltration. Their immunomodulatory activities were compared by studying their immunomodulatory effect on CTX-induced immunosuppressive mice and the release of NO and cytokines from mouse peritoneal macrophages. This method has low sample consumption, a simple experimental process and good reproducibility. This work laid a foundation for clarifying the pharmacodynamics of *CR* polysaccharides with different Mws.

## 2. Materials and Methods

### 2.1. Plant Materials

The experimental plant materials were *C. pilosula* (Franch.) Nannf. from Pingshun of Shanxi Province (No. 1–5, named PSDS), which were identified by Prof. Jianping Gao of Shanxi Medical University.

### 2.2. Instruments

The following instruments were used: Elite liquid chromatograph, differential refractive index detector (RID), Waters 2695 liquid chromatograph, 2489 UV detector, 6000 evaporative light scattering detector (ELSD), CPA225D electronic balance, Neofuge 13R high-speed freezing centrifuge (Shanghai Lishen Scientific Instrument Co., Ltd., Shanghai, China), SC-3610 low-speed centrifuge (Anhui USTC ZONKIA Scientific Instruments Co., Ltd., Hefei, China), ZX-LGJ-18 common freeze dryer (Shanghai Zhixin Experimental Instrument Technology Co., Ltd., Shanghai, China), RE-52 rotary evaporator (Shanghai Yarong Biochemical Instrument Factory, Shanghai, China), cell culture incubator (Li Kang Biomedical Technology Holdings Co., Ltd., Hong Kong, China) and ultrafiltration machine (Shanghai Guxin Biotechnology Co., Ltd., Shanghai, China).

### 2.3. Reagents

Dextran 180, 2500, 4600, 7100, 21,400, 41,100, 84,400, 133,800 and 2,000,000 were obtained from the National Institutes for Food and Drug Control (Beijing, China). Reference compounds, including mannose (Man), glucuronic acid, rhamnose (Rha), galacturonic acid (GalA), glucose (Glc), galactose (Gal), arabinose (Ara), fucose (Fuc) and fructose (Fru) were acquired from Shanxi Jiujiu Trading Co., Ltd. (Taiyuan, China). Trifluoroacetic acid was obtained from China Shanghai Aladdin Biotechnology Co., Ltd. (Shanghai, China), and 1-phenyl-3-methyl-5-pyrazolone was bought from Sinopharm Chemical Reagent Co., Ltd. (Shanghai, China). Analytical methanol, anhydrous ethanol, chloroform and *n*-butanol were purchased from Damao Chemical Reagent Factory (Tianjin, China). Chromatographic-grade acetonitrile was obtained from Thermo Fisher (Thermo, Waltham, MA, USA). RPMI-1640 medium was supplied by Cellgro, Lincoln, NE, USA. Mouse spleen lymphocytes and mouse peritoneal macrophages were provided by Wuhan Psinuo Life Technology Co., Ltd. (Wuhan, China). Lipopolysaccharide (LPS), concanavalin (ConA), MTT and neutral red reagent were purchased from Solarbio (USA). Mouse ELISA kits for IL-2, IL-4, IFN-γ, NO, IL-6, IL-10 and TNF-α were acquired from Boster Biological Technology Co., Ltd. (Wuhan, China).

### 2.4. Preparation and Purification of PSDSs

*CR* raw materials were crushed and passed through a 40-mesh sieve, and then 2.0 g of *CR* powder was placed in a 250 mL round bottom flask containing 60 mL of anhydrous ethanol. The sample was refluxed at 80 °C for 2 h to degrease, and ethanol was recovered under reduced pressure. The residue was soaked in 50 mL of water (material-to-liquid ratio of 1:25) for 20 min and then extracted twice at 95 °C for 1.5 h. Afterwards, the sample was filtered, concentrated to 10 mL, cooled and added with anhydrous ethanol to adjust the alcohol content to 90%. After being left to stand for 12 h, the sample was centrifuged at 3000 r/min for 15 min, and the obtained precipitate was the crude polysaccharide of *CR*. The crude polysaccharide was dissolved in 20 mL of distilled water, mixed with 20 mL of Sevage reagent (chloroform:*n*-butanol = 4:1) in a centrifuge tube, repeatedly shaken for 20–30 min and centrifuged. The supernatant was mixed with 5 mL of Sevage reagent, and the operation was repeated three times to remove protein. The aqueous phase was collected and lyophilised to obtain purified *CR* polysaccharide powder for further use. Yield was calculated using the weight ratio of freeze-dried powder and dried *CR* powder.

### 2.5. Preparation of PSDSs with Different Mws

PSDSs could be grouped into two fractions according to the chromatogram of their Mw distribution (Figure 1A). The Mw of the first fraction is greater than 2000 kDa (out of the linear range), and that of the second fraction is approximately 3.3 kDa. The PSDSs were formulated into a solution of 5 mg/mL and then divided into <30 kDa (PSDSs-1) and >30 kDa (PSDSs-2) using an ultrafiltration membrane with a molecular retention of 100 kDa in the ultrafiltration system. After interception, the two fractions were freeze dried separately.

### 2.6. Physicochemical Properties of PSDSs with Different Mws

#### 2.6.1. Determination of Mw

The relative Mw of PSDSs was determined by high-performance gel permeation chromatography with a TSK gel GMPWXL gel column (300 mm × 7.8 mm, 13 μm, Tosoh, Tokyo, Japan) on an Elite liquid chromatograph with a RID. Purified PSDSs were dissolved in ultrapure water (5 mg/mL) and the injection volume was 10 mL. Ultrapure water was used as the flow phase at a flow rate of 0.8 mL/min under 35 °C. Different Mws of dextran standards were used to calibrate the standard curve. The retention time (*t_R_*) of each standard was plotted on the horizontal axis and lg*M_w_* was plotted on the vertical axis. The relationship between lg*M_w_* and *t_R_* was calculated as shown in the following equation: log*M*_w_ = −0.867*t_R_* + 13.302 (R^2^ = 0.9925).

#### 2.6.2. Determination of Monosaccharide Composition

The monosaccharide composition of PSDSs was determined via acid degradation of polysaccharides combined with pre-column derivatisation. Purified PSDS powder (10 mg) and 6 mL of 2 mol/L tri-fluoroacetic acid solution were added to a plug glass tube. The samples were hydrolysed at 110 °C for 3 h, then cooled and dried under reduced pressure. Briefly, 2 mL of methanol was added and evaporated to dryness, and the operation was repeated three times to remove residual trifluoroacetic acid. Finally, the hydrolysate was dissolved in 1 mL of ultrapure water, and the polysaccharide samples were then derivatised. Approximately 0.2 mL of the solution was mixed with 0.24 mL of 0.5 mol/L 1-phenyl-3-methyl-5-pyrazolone and 0.2 mL of 0.3 mol/L NaOH solution. The solution was placed in a constant temperature metal bath at 70 °C and reacted for 70 min at 300 r/min. Exactly 0.2 mL of 0.3 mol/L HCl was added for neutralisation. Approximately 1 mL of chloroform was added for extraction, and this step was repeated three times to obtain an upper layer of water. Seven monosaccharide standards (Man, Rha, GalA, Glu, Gal, Ara and Fuc) precisely weighed at 0.0225, 0.0228, 0.0265, 0.0495, 0.0225, 0.0751 and 0.0410 g, respectively, were dissolved in 5 mL of ultrapure water. Afterwards, 1 mL of each solution and 4 mL of ultrapure water were mixed to obtain the reference substance solution. The samples were then prepared in accordance with ‘sample derivatisation’. All derivatives were passed through a 0.45 μm microporous membrane.

The derivatives were analysed using a Waters 2695 liquid chromatograph with a 2489 UV detector by a Venusil XBP C18 column (250 mm × 4.6 mm, 5 μm; China Tianjin Bona Agela Technologies Co., Ltd., Tianjin, China). The column was eluted with sodium dihydrogen phosphate buffer (pH 6.7, 50 mmol/L) and acetonitrile (82:18, *v*/*v*) at a flow rate of 1.0 mL/min. The column temperature was set at 35 °C, the chromatograms were monitored at 250 nm and the sample injection volume was 20 µL.

Reference substances Fru and Glc were also prepared as 0.2 mg/mL solution, as previously described [10], and passed through a 0.45 μm microporous membrane. The contents of Fru and Glc in *CR* polysaccharide hydrolysates were determined using a Waters 2695 liquid chromatograph with a 6000 ELSD by an Inertsil NH_2_ column (4.6 mm × 250 mm, 5 μm; Tosoh, Tokyo, Japan). The column was eluted with acetonitrile and water (82:18, *v*/*v*) at a flow rate of 1.0 mL/min. The column temperature was 35 °C and the sample injection volume was 20 µL. The ELSD drift tube temperature was set at 105 °C and the carrier gas was air at a flow rate of 2.5 L/min.

### 2.7. Screening Experiment of Cell Immunological Activity In Vitro

#### 2.7.1. Cell Culture

RAW264.7 cells were cultured in high-Glc DMEM medium supplemented with 10% foetal bovine serum, 100 g/L penicillin and 100g/L streptomycin at 37 °C and 5% CO_2_ saturated humidity. The experiment was carried out after the cells had grown to the logarithmic phase.

#### 2.7.2. Polysaccharide Solution Preparation

PSDSs with different Mws were accurately prepared with concentrations of 25, 50 and 100 μg/mL. A microporous membrane with a diameter of 0.22 μm was used to filter and degerm the solutions for later use.

#### 2.7.3. NO and Cytokine Assay

The levels of NO and cytokines were determined using the Griess and ELISA methods in accordance with the manufacturer’s instructions. RAW264.7 cells were loaded onto a 96-well plate at the density of 2 × 10^5^ cells/mL for 24 h. The cell supernatant was removed and then stimulated with different polysaccharide solutions, LPS (2 μg/mL) or complete DMEM medium for another 24 h. The supernatant was collected and added with 50 μL of Griess Reagents I and II, followed by the measurement of absorbance at 540 nm using a microplate reader. The levels of TNF-α, IL-6 and IL-10 were measured using ELISA kits (Boster Biological Technology Co., Ltd.,Wuhan, China) in accordance with the manufacturer’s instructions.

### 2.8. Screening Experiment of Cell Immunological Activity In Vivo

#### 2.8.1. Experiment Animals and Drug Administration

All animal procedures were performed in accordance with National Institutes of Health Guide for the Care and Use of Laboratory Animals and approved by the Animal Ethics Committee of Shanxi University. Maximum efforts were dedicated to minimising animal suffering and the number of animals necessary for the capture of reliable data.

SPF male ICR mice (18–22 g) were purchased from Beijing Vital River Laboratory Animal Technology Co., Ltd. (Beijing, China) and housed in cages under the experimental conditions with room temperature at 23 ± 1.5 °C, relative humidity 45% ± 15% and 12 h light/dark cycle. The adaptation time lasted for 1 week, during which the mice were provided with free drinking and feeding without any intervention.

After acclimatisation, the mice were randomly divided into eight groups according to their weight (*n* = 8): control group (C), model group (M), PSDSs-1 low-dose group (48 mg/kg, PSDSs-1L), PSDSs-1 middle-dose group (96 mg/kg, PSDSs-1M), PSDSs-1 high-dose group (192 mg/kg, PSDSs-1H), PSDSs-2 low-dose group (2 mg/kg, PSDSs-2L), PSDSs-2 middle-dose group (4 mg/kg, PSDSs-2M) and PSDSs-2 high-dose group (8 mg/kg, PSDSs-2H). From the first day of the experiment, each group was intragastrically administered with the treatment once daily at 0.1 mL/10 g body weight. The mice control group and model group were given sterile water until the end of the experiment. On the 9th day after administration, all the groups (except for the control group) were intraperitoneally injected with CTX at 75 mg/kg for 3 consecutive days.

#### 2.8.2. Effect of PSDSs on the Body Weight of Immunosuppressed Mice

During the experiment, the signs of mind, spirit, appetite and activity of the mice after intervention were observed every day, and their weight was measured every 2 days.

#### 2.8.3. Blood Routine Testing

After the last intragastric administration, 200 μL of orbital blood was taken from each mouse and analysed using a blood cell analyser.

#### 2.8.4. Immune Organ Index

On the 21st day, the animals were sacrificed by cervical dislocation. The spleen and thymus were aseptically separated, washed with normal saline, dried with filter paper, weighed and recorded. The indexes for the spleen and thymus gland were calculated using the following formulas:Spleen index = Weight of spleen (mg)/Weight of the body (g)(1)
Thymus index = Weight of thymus gland (mg)/Weight of the body (g)(2)

#### 2.8.5. Proliferation Assay of Spleen Lymphocytes In Vivo

The mouse spleen cell suspension was prepared as previously described [11] and inoculated into a 96-well culture plate at 2 × 10^6^·mL^−1^ with a volume of 100 μL per well. Different concentrations of each Mw polysaccharide, ConA (final concentration 5 μg·mL^−1^) and LPS (final concentration 10 μg·mL^−1^) were added at 100 μL per well, and six parallel wells were established. The culture plate was placed in a 37 °C and 5% CO_2_ incubator for 48 h. 10 μL of MTT was then added to each well and the culture was continued for 4 h. The culture medium was discarded by centrifugation and 200 μL of DMSO was added to each well. The sample was shaken at a low speed on a shaker to fully dissolve the crystals. The absorbance value (A value) was detected at 570 nm using a microplate reader.

#### 2.8.6. Determination of IL-2, IL-4 and IFN-γ Secreted by Spleen Lymphocytes

The spleen cells and ConA were co-cultured in RPMI-1640, and the supernatant was obtained. The contents of IL-2, IL-4 and IFN-**γ** were determined by ELISA in accordance with the kit instructions. The OD value of the culture solution at a wavelength of 450 nm was determined using a microplate reader. The corresponding content was determined from the cytokine standard curve.

#### 2.8.7. Neutral Red Uptake by Peritoneal Macrophages In Vivo

The mouse peritoneal macrophage suspension was prepared as previously described [12] and inoculated into a 96-well plate at 1 × 10^6^·mL^−1^. The volume of each well was 200 μL, and six parallel wells were established. The 96-well plate was placed in a 37 °C and 5% CO_2_ incubator for 2 h. The culture medium was discarded, the wells were washed three times with PBS and then 200 μL of different concentrations of polysaccharides with various Mws were added. After being cultured for 24 h, 200 μL of 0.075% polysaccharide was added to each well. Neutral red physiological saline solution was added to the wells, cultured for 1 h and then discarded. The cells were washed three times with PBS, then 200 μL of cell lysis buffer (acetic acid:ethanol = 1:1) was added and allowed to stand overnight. The absorbance value (A value) was measured at 540 nm with a microplate reader.

#### 2.8.8. Spleen NK Cell Activity Assay In Vivo

The mouse spleen cell suspension was prepared as previously described [13]. Mouse spleen lymphocytes were used as effector cells, and YAC-1 cells were used as target cells. In a 96-well culture plate, different concentrations of each polysaccharide were mixed with the target cells in a culture medium acting as the natural release hole. Meanwhile, different concentrations of each polysaccharide were mixed with the target cells in 1% NP40 used as the maximum release hole. Polysaccharides, target cells and NK cells (mouse spleen lymphocyte fluid) were used as reaction wells. Six parallel wells were established. The plate was incubated in a 37 °C and 5% CO_2_ incubator for 44 h. After centrifugation, 100 μL of supernatant was aspirated from each well and placed in a culture plate. The obtained supernatant was added with 100 μL of LDH matrix solution and allowed to react for 8 min. 30 μL of 1 mol·L-1 HCl was then added to each well, and the absorbance value (A value) was measured at 490 nm using a microplate reader.

### 2.9. Statistical Analysis

All experimental data were expressed as mean ± standard deviation (x ± s) and statistically analysed using SPSS 16.0. A T-test was used to determine the significance between groups, and *p* < 0.05 indicated statistical significance.

## 3. Results and Discussion

### 3.1. Physicochemical Properties of PSDSs with Different Mws

#### 3.1.1. Determination of the Yield of PSDSs

The average yield of PSDSs was 25.09% (23.45–26.37%).

#### 3.1.2. Determination of the Mw Distribution of PSDSs

HPLC was used to analyse the Mw distribution of the five batches of PSDSs, and the overlapping results of the samples are shown in Figure 1A. The chromatographic peak in Figure 1A represents the relative Mw at the highest point of each cutting part.

As shown in Figure 1A, the Mw distribution of PSDSs could be divided into two fractions: the Mw of the first fraction is greater than 2000 kDa (out of the linear range), and that of the second fraction is approximately 3.3 kDa. The peak area of each fraction differed in the percentage of the total peak area. As shown in Figure 1B, the polysaccharides with a Mw of approximately 3.3 kDa accounted for approximately 96% of the total polysaccharides. Meanwhile, those with a Mw greater than 2000 kDa accounted for approximately 4%.

Li et al. [14] found that the fraction with a Mw of approximately 3 kDa consists of the main component, namely, *CR* polysaccharides which has anti-gastric ulcer activity and immunomodulatory effects. Li et al. [15] also found that *CR* polysaccharides with a Mw of 3 kDa (fraction 2) have a significant, time-dependent growth-stimulating effect on *Bifidobacterium longum* and could play a prebiotic effect. All these results showed that the fraction with a Mw of approximately 3 kDa is the main component that *CR* polysaccharide take effect. The Mw of approximately 3.3 kDa in PSDSs accounted for roughly 96% of the total polysaccharide, reflecting the characteristics of Shanxi Pingshun ludangshen as an authentic medicinal material.

#### 3.1.3. Determination of the Monosaccharide Composition of PSDSs

Most current detectors could not recognise monosaccharides because their molecular structure lacks chromogenic functional groups. Derivatisation could add UV or fluorescent groups to sugar chains to improve the detection sensitivity. The commonly used derivatisation reagents are UV and fluorescent. Among the many derivatisation reagents, the reaction between 1,3-substituted pyrazolone derivatisation reagents represented by 1-phenyl-3-methyl-5-pyrazolone and the reducing end of the sugar chain can be carried out in a weakly alkaline medium. Compared with other derivatisation reagents that require an acidic medium, 1-phenyl-3-methyl-5-pyrazolone has the advantages of mild conditions, stable derivatised products, no stereoisomers and strong UV absorption. It is widely used in the analysis of carbohydrates [16]. Fructose is a ketose that is difficult to derivatise due to the lack of a reducing end and could not be detected by a UV detector. ELSD is a general purpose mass detector that could be used for the determination of (Fru) ketose due to its ability to measuring compounds without UV absorption [17]. Therefore, this experiment used HPLC-ELSD combined with HPLC-UV to determine the monosaccharide composition of PSDSs.

HPLC-ELSD and HPLC-UV were used for the characteristic mapping of eight standard monosaccharide mixtures (Figure 2A,C) and PSDS monosaccharides (Figure 2B,D). Comparison showed that the PSDSs were composed of six monosaccharides, namely, Rha, GalA, Glc, Gal, Ara and Fru. With the peak area in the monosaccharide composition determination result taken as an index, the peak area of glucose was set as a unit of 1.00, the peak area of other monosaccharides was compared with that occupied by glucose and the relative content ratio between various polysaccharides was calculated. For rhamnose:galacturonic acid:glucose:galactose:arabinose:fructose = 0.05:0.08:1.00:0.13:0.13:1.24. When the amount of Glc was 1.00, the ratios were nearly 0.05:0.08:1.00:0.13:0.13:1.24.

Zhang et al. [10] analysed the monosaccharide composition of *CR* polysaccharides from base sources and growing areas and found that the main monosaccharides of *CR* polysaccharides are fructose and Glc and the most abundant is Fru. This finding was consistent with the current experimental results.

The composition of monosaccharides is closely related to the chain structure and advanced structure of polysaccharides, so a change in monosaccharide composition will lead to different immune functions or other biological activities. Wang [9] showed that the polysaccharide with a relative Mw of 1698 Da has a good immunomodulatory activity and has glucose as a monosaccharide component. Therefore, the existence of glucose in monosaccharide composition is associated with a good immunomodulatory effect. Meng et al. [8] studied the anti-inflammatory activities of two kinds of Codonopsis polysaccharides with different Mws and found that CP1-2-1 composed of Fru and CP3-1-1 composed of Ara, Rha, Gal and GalA both have certain anti-inflammatory activities. The mechanism may be related to the inhibited activation of TLR4/NF-κB pathway to suppress the generation and release of inflammatory factors, such as TNF-α and IL-6. The probiotic activity of polysaccharides is strongly associated with their monosaccharide composition. The different monosaccharide compositions of polysaccharides may lead to different effects on intestinal flora, which is closely related to the secretion and expression of cytokines in the body and interacts to regulate immune function. Huang et al. [18] showed that most of the polysaccharides that regulate the activity of intestinal flora have Man, Glc, Rha, Gal and Ara; the activity of probiotics is strengthened when the Glc content is high. In the present work, the monosaccharide composition of PSDSs comprised Rha, GalA, Glc, Gal, Ara and Fru. This finding was consistent with the above literature; therefore, PSDSs have good immunoregulatory activity.

### 3.2. Effects of PSDSs with Different Mws on the Immune System In Vitro

#### 3.2.1. Effects of PSDSs on LPS-Induced NO Secretion of RAW264.7 Macrophages In Vitro

Figure 3A shows that the content of NO in the LPS group increased significantly compared with that in the control group (*p* < 0.001). Meanwhile, high-dose PSDSs-1 significantly inhibited NO production compared with that in the LPS group (*p* < 0.01). These results showed that PSDSs-1 help promote the inflammatory response of LPS-stimulated macrophages and show potential as an anti-inflammatory drug.

#### 3.2.2. Effects of PSDSs on LPS-induced IL-6, TNF-α and IL-10 Secretion of RAW264.7 Macrophages In Vitro

Figure 3B,C show that LPS significantly stimulated the production of TNF-α and IL-6 compared to the control group. PSDSs-1 significantly promoted the production of TNF-α and IL-6, and high-dose PSDSs-2 significantly inhibited the production of TNF-α and IL-6. Figure 3D shows that, compared to the LPS group, PSDSs-1 significantly inhibited the IL-10 secretion of RAW264.7 cells in a concentration-dependent manner at 25–100 μg/mL and PSDSs-2 significantly promoted the IL-10 secretion of RAW264.7 cells. TNF-α and IL-6 are pro-inflammatory cytokines, and IL-10 is anti-inflammatory factor. The above experimental results showed that PSDSs-1 have pro-inflammatory effect, and PSDSs-2 have anti-inflammatory effect.

Macrophages, which have the functions of phagocytosis, antigen presentation and secretion of various bioactive substances, are one of the most important immune cells in the body. When acute inflammatory reaction occurs, a large amount of NO will be produced by macrophages, resulting in cell damage. The degree of inflammation can be reflected by measuring the content of NO in cell culture medium. The increase of the secretion of inflammatory factors (such as IL-6 and TNF-α) will lead to inflammation. Many factors can prompt cells to secrete additional inflammatory factors. This study showed that after LPS treatment, the release of NO in inflammatory model cells was significantly increased and the levels of IL-6, TNF-α and IL-10 were increased, indicating that the inflammatory model was successfully established. After the treatment of PSDSs with different Mws, the NO release of model group was significantly decreased. Compared with those in the model group, the levels of IL-6 and TNF-α increased and the level of IL-10 decreased after PSDSs-1 was given. After PSDSs-2 was given, the levels of IL-6 and TNF-α decreased and the level of IL-10 increased. These experimental results showed that PSDSs-1 have a certain pro-inflammatory activity, and PSDSs-2 have a certain anti-inflammatory activity. The mechanism may be related to the activation of the TLR4/NF-κB pathway to regulate the production and release of inflammatory factors, such as TNF-α and IL-6.

### 3.3. Effects of PSDSs with Different Mws on the Immune System In Vivo

#### 3.3.1. Effect of PSDSs with Different Mws on the Body Weight of Immunosuppressed Mice

Figure 4A shows that before CTX injection, the weight of each group increased with time. Particularly, the weight gain of the treatment groups was higher than that of the control group and the model group. After CTX injection, the body weight decreased for all groups except the control group. However, the body weight of each treatment group was lower than that of the model group. Among the treatment groups, those treated with PSDSs-1 had the closest body weight to the control group. The body weight of the high-dose group was significantly higher than that of the middle- and low-dose groups.

#### 3.3.2. Effect of PSDSs with Different Mws on the Routine Haematological Parameters of Immunosuppressed Mice

After the last intragastric administration, the blood routine test results of each group of mice are shown in Table 1. The blood routine test can confirm whether the mouse model was successfully established. Blood routine indicators include the number of white blood cells, lymphocytes, red blood cells, haemoglobin and platelets. As shown in Table 1, the blood routine indicator levels of the model group were significantly lower than those of the control group (*p* < 0.01). This finding indicates the successful establishment of the model. Additionally, PSDSs-1 can significantly affect the level of blood routine indicators.

#### 3.3.3. Effect of PSDSs with Different Mws on the Immune Organ Index of Mice

The thymus and spleen are the main immune organs of the body, and their indexes reflect the non-specific immunity of the body [19]. Therefore, the effect of drugs on the thymus and spleen indexes can be used as a preliminary parameter to study the immune pharmacological mechanisms in an animal model [20]. Cyclophosphamide inhibits lymphocyte differentiation and reduces the number of lymphocytes in immune organs, resulting in a decrease of the weight of the spleen, thymus and body [21]. As shown in Table 2, the spleen and thymus indexes of the model group were significantly lower than those of the control group (*p* < 0.01), indicating that CTX had a certain damaging effect on the immune organs of mice. Compared with those in the model groups, the spleen and thymus indexes increased to different degrees in the treatment groups, and the middle and high doses of PSDSs-1 exerted significant effects on the immune organ index (*p* < 0.05). These results showed that PSDSs-1 could improve the organ damage caused by CTX, promote the development of immune organs in immunosuppressed mice and enhance their immunity.

#### 3.3.4. Effect of PSDSs with Different Mws on the Proliferation of Mouse Spleen Lymphocytes In Vivo

As shown in Figure 5A,B, the proliferation of spleen lymphocytes in the model group was significantly lower than that in the control group (*p* < 0.001), indicating that CTX can inhibit the specific immune function of normal mice. Compared with that in the model group, the spleen lymphocyte proliferation of mice in each treatment group increased by varying degrees, and the effect of high-dose PSDSs-1 was the most significant (*p* < 0.001). This finding indicated that PSDSs-1 may play a specific immune function by promoting the proliferation of splenic lymphocytes in mice.

#### 3.3.5. Effect of PSDSs with Different Mws on the IL-2, IL-4 and IFN-γ Secretion of Mouse Spleen Lymphocytes

When spleen lymphocytes are activated, T lymphocytes secrete cytokines, including IL-2, IL-4 and IFN-**γ,** which are important for acute inflammatory reaction and immune enhancement. The role of IL-2 and IL-4 is to resist viral infection, activate T cells, promote B cell proliferation and secrete antibodies. IFN-**γ** can inhibit viruses, eliminate bacteria and enhance the immune activity of macrophages and NK cells. As shown in Figure 5C–E, the spleen lymphocyte cytokine level in the model group was significantly lower than that in the control group (*p* < 0.0001). Compared with that in the model group, the cytokine level in each treatment group increased by different degrees, and the effect of high-dose PSDSs-1 was the most significant (*p* < 0.0001). Meanwhile, PSDSs-2 had a minimal effect on the secretion of IL-2, IL-4 and IFN-γ in mouse spleen lymphocytes. These results showed that PSDSs-1 play the most important role in the two parts of specific immunity.

#### 3.3.6. Effect of PSDSs with Different Mws on the Phagocytic Activity of Mouse Peritoneal Macrophages In Vivo

The phagocytosis of macrophages plays an important role in the immunological response and contributes to the immune function in animals. As shown in Figure 5F, the phagocytic function of the model group was significantly lower than that of the control group (*p* < 0.0001), indicating that CTX had a certain damaging effect on the non-specific immune function of mice. Compared with that in the model group, the phagocytosis in each treatment group was enhanced by varying degrees, and the effect of high-dose PSDSs-1 was the most significant (*p* < 0.0001). Meanwhile, PSDSs-2 had a minimal effect on the phagocytic activity of peritoneal macrophages in mice.

#### 3.3.7. Effect of PSDSs with Different Mws on the Activity of Spleen NK Cells In Vivo

As shown in Figure 5G, the NK cell activity of spleen in model group was significantly lower than that in control group (*p* < 0.0001). Compared with that in the model group, the spleen NK cell activity of each treatment group increased by different degrees, and high-dose PSDSs-1 had the most significant effect (*p* < 0.001). These results showed that PSDSs-1 play a role in non-specific immunity by increasing NK cell activity in the spleen.

*CR* polysaccharides are an important natural active ingredient in *C. pilosula* and an important signal molecule in vivo. They have a strong regulatory effect on non-specific immunity and specific immunity. As an intuitive manifestation of any disease, body weight can simply and clearly show the effectiveness of drugs. In this study, high-dose PSDSs-1 showed the strongest effect on the body weight of mice.

Macrophages are an important cellular component of the body’s immune system and play an extremely important role in the innate and adaptive immune functions of the body. They can phagocytose and kill pathogens and foreign bodies, absorb and process antigens and provide antigenic information to lymphocytes. They can release a variety of cytokines after activation. Cytokines can be divided into interleukin, interferon and tumour necrosis factor according to their functions. The effect of polysaccharides on the function of activated NK cells is also an important aspect in evaluating the non-specific immune enhancement activity of these molecules. The activation of immune response by macrophages and NK cells is one of the important mechanisms of polysaccharides with known immune activity. Macrophages and NK cells are the main immune cells involved in non-specific immunity. This study showed that PSDSs with different Mws could stimulate macrophages to produce neutral red and NK cells to proliferate in varying degrees, suggesting that PSDSs could enhance non-specific immune response by stimulating macrophages and NK cells. Shi et al. [22] reported that *CR* polysaccharides can significantly increase the secretion of IL-1β and TNF-α of macrophage Ana-1, significantly enhance the proliferation of T lymphocytes and improve the thymus index of experimental mice and the content of TNF-α and IL-1β in serum in a dose-dependent manner. Shi et al. [23] also found that *CR* polysaccharides could increase the secretion of TNF-α and IL-6 by stimulating the proliferation of RAW 264.7 cells and activating the NF-кB signalling pathway, thereby participating in the immune regulation of the body. Wang [24] found that PSDSs can stimulate macrophages, thereby increasing the amount of TNF-α, IL-6, IL-1β and IL-10 secreted by macrophages. Real-time fluorescence quantitative results also showed that macrophages could increase the expression levels of the above cytokines after being activated by PSDSs. Yin [25] found that PSDSs can increase the levels of IFN-γ, IL-2, IL-4 and IL-6 in the serum of piglets, and the effect of 2% PSDSs is better than that of 1% PSDSs.

As important immune organs, thymus and spleen indexes can reflect the strength of the immune function. Lymphocytes come from the spleen. Con A is a mouse T cell-specific mitogen. The increase of polysaccharide and Con A content can promote the proliferation of mouse lymphocytes, reflecting the effect of polysaccharide on T lymphocyte proliferation. LPS is a commonly used B cell activator that can induce the proliferation and differentiation of B lymphocytes. PSDSs-1 could promote LPS-induced splenic lymphocyte proliferation in coordination with LPS, suggesting that PSDSs-1 could promote B lymphocyte differentiation. PSDSs with different Mws can stimulate the specific immune system by promoting lymphocyte proliferation and secreting cytokines. Zhang et al. [26] used tetramethylazolium salt trace enzyme colorimetry (MTT) to study the effect of *C. pilosula* polysaccharide on the proliferation of B lymphocytes in mice and found that *C. pilosula* could stimulate the proliferation of spleen B lymphocytes in a dose-dependent manner. Xu et al. [27] found that PSDSs in varying doses can improve the thymus and spleen indexes of immunocompromised mice, thereby enhancing their immune function.

The above results showed that PSDSs-1 have better immune enhancement effect than PSDSs-2 and can influence specific immunity and non-specific immunity, cellular immunity and humoral immunity. This activity may be related to the low Mw of PSDSs-1. Generally, low Mw polysaccharides have high biological activity. A small Mw is correlated with a small molecular volume which is conducive to the polysaccharide crossing multiple membrane barriers into the organism to play its biological activity.

In vitro inflammation model experiments showed that PSDSs-1 exhibited pro-inflammatory activity and PSDSs-2 had the opposite function and exhibited anti-inflammatory activity, indicating the complexity of the immunomodulatory activity of PSDSs with different Mws. In the in vivo test, the immunosuppressive mouse model of cyclophosphamide was used to study the activity of PSDSs. The findings mainly reflected the immune recovery or immune enhancement function of PSDSs with different Mws and verified that the immune enhancement activity of PSDSS-1 was better than that of PSDSs-2. However, a large number of research results showed that PSDSs have a significant effect on immune promotion and a certain anti-inflammatory activity [8,14,28,29]. Our research group is currently using the ulcerative colitis animal model for follow-up verification to further screen out the material basis of the anti-inflammatory components of PSDSs with different Mws. On the basis of the study of two different animal models (cyclophosphamide immunosuppressive model and inflammatory animal model) and the structural characteristics of PSDSs with different Mws, the present work established the structure–activity relationship of PSDSs and laid a foundation for their further classification, development and application.

On the basis of the experimental results and related literature, the mechanism of immune regulation by PSDSs may be related to the secretion of inflammatory factors caused by the proliferation of macrophages and NK cells, the improvement of immune organ function, the proliferation of lymphocytes and the activation of the NF-κB signalling pathway.

## 4. Conclusions

*CR* is a common Chinese medicine that nourishes the blood and body fluid and invigorates the spleen and lungs. It is cultivated in many places in China. Traditional Chinese medicine is a complex multi-component system. The formation, transformation and accumulation of its active ingredients are affected by the growth of the external environment. Polysaccharide is an important component of *CR*, affecting its ability to exert its immunomodulatory effect, and thus, should be considered as an identification and evaluation index for *CR*.

The Mw distribution of PSDSs could be divided into two fractions: those with a Mw greater than 2000 kDa (out of the linear range) (PSDSs-2) (and those with a Mw of approximately 3.3 kDa (PSDSS-1). The Mw peak areas of these fractions accounted for 4% (PSDSS-2) and 96% (PSDSS-1) of the total peak area. PSDSs are composed of Rha, GalA, Glc, Gal, Ara and Fru. On the basis of its monosaccharide composition, PSDs were speculated to have good immunoregulatory activity.

PSDSs-1 have a pro-inflammatory effect, and PSDSs-2 exhibit a certain anti-inflammatory activity. The mechanism involved may be related to the activation of the TLR4/NF-κB pathway to regulate the production and release of inflammatory factors, such as TNF-α and IL-6. Additionally, PSDSs-1 have significant aboriginal immune enhancement activity, and this finding was consistent with the above speculation regarding immune activity based on monosaccharide composition. The mechanism may be related to the secretion of inflammatory factors caused by the proliferation of macrophages and NK cells, the improvement of the function of immune organs, the proliferation of lymphocytes and the activation of the NF-кB signalling pathway.

In this work, the polysaccharides of different Mws of *C. pilosula* were separated for the first time by ultrafiltration. The immunological activities of PSDSs with different Mws were studied in vivo and in vitro, and the structure and activity of *CR* were analysed comprehensively. However, this study also has certain limitations. This research only analysed the monosaccharide composition of *CR* total polysaccharides and did not conduct in-depth monosaccharide composition and structural characterisation studies on the isolated polysaccharides of different Mws. The established characteristic maps of polysaccharides and monosaccharides combined with the evaluation method of immune activity provide a new research idea for the quality control of Chinese herbal medicine polysaccharides. Meanwhile, this study serves as a reference for further research on PSDSs with different structures and immune activities, and provides guidance for the quality control of PSDSs and the development of new products. We will also supplement the shortcomings of this paper in future experiments and further clarify the specific structures of PSDSs with different Mws and the specific reasons underlying the differences in immune activity caused by structural differences.

## Figures and Tables

**Figure 1 molecules-27-05454-f001:**
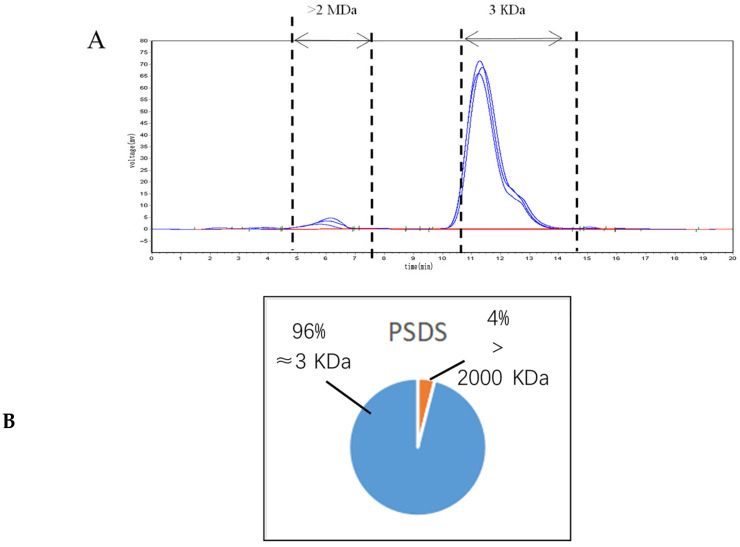
HPLC-RID chromatogram of purified polysaccharide from PSDS (**A**); Molecular weight peak area of each fraction of *CR* polysaccharides accounting for the percentage of total peak area (**B**).

**Figure 2 molecules-27-05454-f002:**
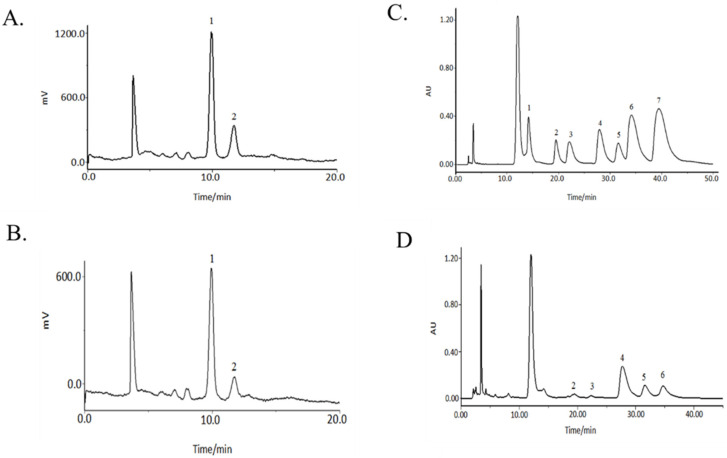
Monosaccharide chromatograms of reference substances (**A**) and PSDSs (**B**) by HPLC-ELSD; monosaccharide chromatograms of reference substances (**C**) and PSDSs (**D**) by HPLC-UV. (**A**,**B**) (1: Fru; 2: Glc); (**C**,**D**) (1: Man; 2: Rha; 3: GalA; 4: Glc; 5: Gal; 6: Ara; 7: Fuc).

**Figure 3 molecules-27-05454-f003:**
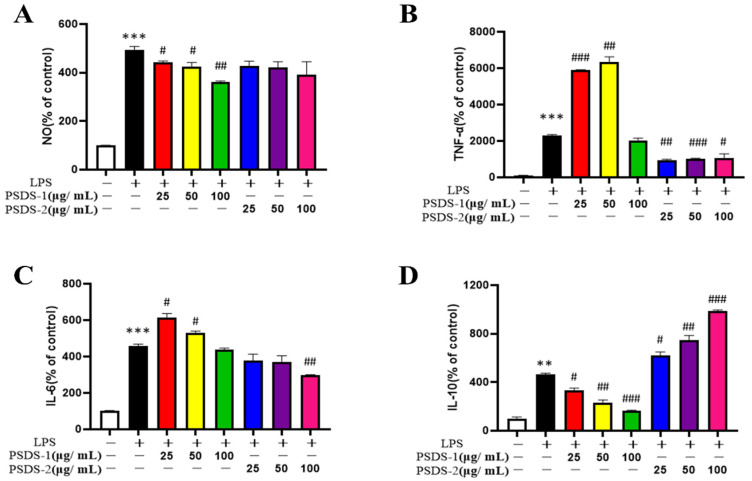
Effects of PSDSs-1 and PSDSs-2 on NO release in RAW264.7 cells (**A**); Effects of PSDS-1 and PSDSs-2 on (**B**) TNF-α production, (**C**) IL-6 production and (**D**) IL-10 production in RAW264.7 cells. ** *p* < 0.01, *** *p* < 0.001 compared with blank control group; ^#^ *p* < 0.05, ^##^ *p* < 0.01 and ^###^ *p* < 0.001 compared with LPS.

**Figure 4 molecules-27-05454-f004:**
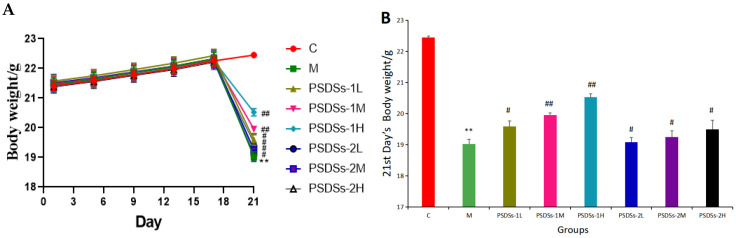
Effects of PSDSs of different Mw on the body weight of mice (**A**); Effects of PSDSs of different Mw on the 21st day body weight of mice (**B**). *n* = 8, x ± s. ** *p* < 0.01 vs. blank control group; ^#^ *p* < 0.05, ^##^ *p* < 0.01 vs. model group.

**Figure 5 molecules-27-05454-f005:**
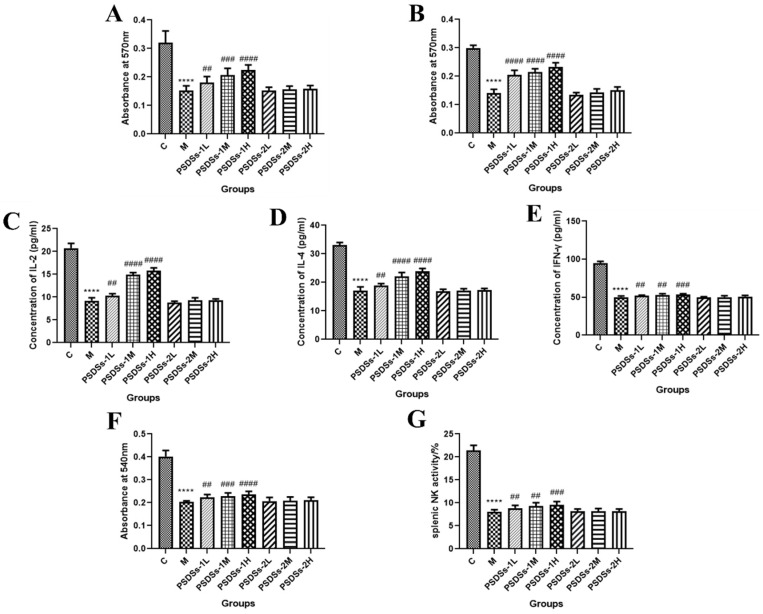
Effects of PSDSs on the proliferation of lymphocytes combined with ConA (**A**) or LPS (**B**). Effect of PSDSs on the secretion of IL-2 (**C**), IL-4 (**D**) and IFN-γ (**E**) induced by spleen lymphocytes in immunosuppressed mice. Effects of PSDSs on the phagocytic activity of peritoneal macrophages (**F**). Effects of PSDSs on splenic NK activity (**G**). *n* = 8, x ± s. **** *p* < 0.0001 vs. blank control group; ^##^ *p* < 0.01, ^###^ *p* < 0.001, ^####^ *p* < 0.0001 vs. model group.

**Table 1 molecules-27-05454-t001:** The effect of PSDSs with different Mw on routine hematological parameters in mice.

Groups	White Blood Cell	Lymphocyte	Red Blood Cell	Haemoglobin	Blood Platelet
	WBC (×10^9^/L)	Lym (×10^9^/L)	RBCs (×10^12^/L)	HGB (g/L)	PLT (×10^9^/L)
C	4.57 ± 1.43	1.88 ± 0.23	9.13 ± 1.63	135 ± 5.80	956 ± 145.11
M	1.91 ± 0.74 ***	0.65 ± 0.15 ****	7.32 ± 0.40 **	120 ± 9.97 **	542 ± 107.22 ****
PSDSs-1L	2.76 ± 0.16 ^##^	0.97 ± 0.33 ^#^	8.98 ± 1.11 ^##^	130 ± 9.19 ^#^	671 ± 118.02 ^#^
PSDSs-1M	2.88 ± 0.17 ^##^	1.03 ± 0.31 ^##^	9.03 ± 1.09 ^###^	132 ± 10.22 ^#^	685 ± 64.36 ^##^
PSDSs-1H	2.99 ± 0.27 ^##^	1.09 ± 0.36 ^##^	9.08 ± 1.10 ^###^	136 ± 5.81 ^##^	701 ± 78.68 ^##^
PSDSs-2L	1.93 ± 0.27	0.72 ± 0.21	7.40 ± 1.22	122 ± 3.46	524 ± 91.81
PSDSs-2M	2.00 ± 0.40	0.70 ± 0.24	7.48 ± 1.67	125 ± 3.30	535 ± 106.83
PSDSs-2H	2.07 ± 0.29	0.75 ± 0.19	7.53 ± 1.53	127 ± 4.14	547 ± 92.40

*n* = 8, x ± s. Compared with control group, ** *p* < 0.01, *** *p* < 0.001, **** *p* < 0.0001; Compared with model group, ^#^ *p* < 0.05, ^#^^#^ *p* < 0.01, ^#^^#^^#^ *p* < 0.001.

**Table 2 molecules-27-05454-t002:** Effect of PSDSs with different Mws on the immune organ index of mice.

Group	Thymus Index (mg/g)	Spleen Index (mg/g)
C	2.86 ± 0.14	4.46 ± 0.05
M	1.35 ± 0.08 **	2.15 ± 0.07 **
PSDSs-1L	1.47 ± 0.07	2.98 ± 0.09
PSDSs-1M	1.52 ± 0.04 ^#^	3.06 ± 0.07 ^#^
PSDSs-1H	1.62 ± 0.05 ^#^	3.11 ± 0.06 ^#^
PSDSs-2L	1.37 ± 0.06	2.49 ± 0.05
PSDSs-2M	1.49 ± 0.09	2.52 ± 0.03
PSDSs-2H	1.41 ± 0.10	2.56 ± 0.07

*n* = 8, x ± s. Compared with control group, ** *p* < 0.01; Compared with model group, ^#^ *p* < 0.05.

## Data Availability

The data presented in this study are available in this article.

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
