# Peer review of "Screening of Codonopsis radix Polysaccharides with Different Molecular Weights and Evaluation of Their Immunomodulatory Activity In Vitro and In Vivo"

_molecules, 2022, doi:10.3390/molecules27175454_

Round 1

Reviewer 1 Report

This article discusses the immunostimulant properties of polysaccharides derived from Codonopsisi radix.

However, several considerations must be taken into account before publication.

1. According to the authors' guide, the summary should not exceed 200 words. There are 446 words in the current abstract. An abstract should present the article objectively: it cannot contain results that are not presented in the main text, nor should it exaggerate its conclusions.

2. Materials and methods could be improved by specifying the techniques used; for example, section 2.8.5 does not specify the technique used to assess lymphocyte proliferation. The experiment scheme is the only thing indicated. While well-established methods can be briefly described and appropriately cited, new methods and protocols should be described in detail.

3. The controls of polysaccharides alone are not shown in figure number 3, as there is ample evidence that polysaccharides of different origins have immunostimulatory and immunoregulatory effects.

4. Figure 4 should include the concentrations used in mice. A figure of histograms based only on the weight data of day 21 would also be interesting, since this would help to show the differences between groups more clearly.

5. While the discussion is presented in conjunction with the results, it was thought that the authors should discuss the results more extensively, and how they can be interpreted from the perspective of previous studies and working hypotheses. Furthermore, they must be supported by bibliographical references, which is lacking in the writing, particularly in the discussion of results.

6. The results of in vitro experiments indicate that these polysaccharides have immunoregulatory activity, while in vivo results indicate that they have immunostimulant activity, which at first glance seems contradictory. However, by getting better at your discussion, this may become clearer.

Author Response

Point 1: According to the authors' guide, the summary should not exceed 200 words. There are 446 words in the current abstract. An abstract should present the article objectively: it cannot contain results that are not presented in the main text, nor should it exaggerate its conclusions.

Response 1: Thank you for your suggestion. The abstract has been limited to 200 words, and the main conclusions in it have been verified to improve the accuracy of the main results in the abstract.

Point 2: Materials and methods could be improved by specifying the techniques used; for example, section 2.8.5 does not specify the technique used to assess lymphocyte proliferation. The experiment scheme is the only thing indicated. While well-established methods can be briefly described and appropriately cited, new methods and protocols should be described in detail.

Response 2: Thanks for your comments, we have made revisions to address such issues, and made targeted supplementary explanations for the content of the materials and methods in the text. Specific modification 2.8.5, 2.8.7, 2.8.8

Point 3: The controls of polysaccharides alone are not shown in figure number 3, as there is ample evidence that polysaccharides of different origins have immunostimulatory and immunoregulatory effects.

Response 3: Thank you for your comments. Figure 3 in this article compares the in vitro immune activities of Codonopsis pilosula polysaccharides with different molecular weights, and the mutual comparison of different polysaccharides has been carried out.

Point 4: Figure 4 should include the concentrations used in mice. A figure of histograms based only on the weight data of day 21 would also be interesting, since this would help to show the differences between groups more clearly.

Response 4: Thank you for your suggestion. This article has added the dosing concentration of mice in each group in the title of the figure, and also added a histogram based on the weight data of mice on the 21st day.

Point 5: While the discussion is presented in conjunction with the results, it was thought that the authors should discuss the results more extensively, and how they can be interpreted from the perspective of previous studies and working hypotheses. Furthermore, they must be supported by bibliographical references, which is lacking in the writing, particularly in the discussion of results

Response 5: Thanks for your suggestion, we have supplemented the discussion of the results in this paper with four references for a more in-depth interpretation of the experimental results based on the literature. For details, see 3.3.7 of this article.

Point 6: The results of in vitro experiments indicate that these polysaccharides have immunoregulatory activity, while in vivo results indicate that they have immunostimulant activity, which at first glance seems contradictory. However, by getting better at your discussion, this may become clearer.

Response 6: Thank you for your valuable comments. As requested, we have conducted a more in-depth analysis of the research results in order to make the functional research of Codonopsis polysaccharide clearer.

In vitro inflammation model experiments showed that PSDSs-1 exhibited pro-inflammatory activity and PSDSs-2 had the opposite function and exhibited anti-inflammatory activity, indicating the complexity of the immunomodulatory activity of PSDSs with different Mws. In the in vivo test, the immunosuppressive mouse model of cyclophosphamide was used to study the activity of PSDSs. The findings mainly reflected the immune recovery or immune enhancement function of PSDSs with different Mws and verified that the immune enhancement activity of PSDSS-1 was better than that of PSDSs-2. However, a large number of research results showed that PSDSs have a significant effect on immune promotion and a certain anti-inflammatory activity[28,29,30,31]. Our research group is currently using the ulcerative colitis animal model for follow-up verification to further screen out the material basis of the anti-inflammatory components of PSDSs with different Mws. On the basis of the study of two different animal models (cyclophosphamide immunosuppressive model and inflammatory animal model) and the structural characteristics of PSDSs with different Mws, the present work established the structure–activity relationship of PSDSs and laid a foundation for their further classification, development and application.

Reviewer 2 Report

The manuscript deals with identification of polysaccharides from Codonopsis Radix and evaluation of their immunomodulatory activity. The topic is of interest and has impact in the field of natural bioactive compounds practical aplication. The manuscript has typical structure and logically presented. A wide range of tests has been applied for the confirmation of polysaccharides properties both in vitro and in vivo. Conclusions are fully supported with experimental data. The manuscript can be accepted to publication after minor revision.

1. English needs revision. There are badly constructed phrases throughout the text.

2. Abbreviations have to be carefully checked. The full description should be presented at the first mention both in the abstract and the main text. Other abbreviations have to be removed as far as used just one-two times, for example HPGPC, TFA,

3. Introduction, several relevant references are missed.

https://doi.org/10.1016/j.biopha.2019.108682

https://doi.org/10.1016/j.ijbiomac.2020.05.083

https://doi.org/10.1080/07328303.2020.1772278

4. Latin words should be italicized throughout the manuscript.

5. Figure 5 quality is too low. High resolution image is required.

Author Response

Point 1: English needs revision. There are badly constructed phrases throughout the text.

Response 1: Thanks for your comments, the full text has been revised and revised by professional English speakers.

Point 2: Abbreviations have to be carefully checked. The full description should be presented at the first mention both in the abstract and the main text. Other abbreviations have to be removed as far as used just one-two times, for example HPGPC, TFA,

Response 2: Thanks for your suggestion, this question has been checked for full text and modified accordingly.

Point 3: Introduction, several relevant references are missed.

https://doi.org/10.1016/j.biopha.2019.108682

https://doi.org/10.1016/j.ijbiomac.2020.05.083

https://doi.org/10.1080/07328303.2020.1772278

Response 3: Thank you for your suggestion, the literature is not in place, and it has been revised for this issue.

Point 4: Latin words should be italicized throughout the manuscript

Response 4: Thanks for your suggestion, these questions have been modified accordingly.

Point 5: Figure 5 quality is too low. High resolution image is required.

Response 5: Thanks for the suggestion, this article has increased the resolution of the full text image for this issue.

Reviewer 3 Report

REVIEWER’S COMMENT

Major revision comments

1.      Abstract – it is not clear what the novelty of this study is. There are some data regarding the percentage of monounsaturated fatty acids. Are these data the IQRs of the medians?

2.      Materials and methods:

Line 169: are the LC methods for the determination of monosaccharide composition worked out in house or they are adopted from another paper with modifications ? How was done the quantitative analysis? This needs to be described in the manuscript.

3.      Results and Discussion:

Line 400-402: Figure 4 – the legend is not clear. What mean M, -1L, -1M, -1H etc.?

The strengths and the limitations of the study should be specified.

Technical issues:

1.         The whole text needs thoroughly editing.

Author Response

Point 1: Abstract – it is not clear what the novelty of this study is. There are some data regarding the percentage of monounsaturated fatty acids. Are these data the IQRs of the medians?

Response 1: Thank you for your suggestion, the Abstract has been revised in response to the suggestion, highlighting the novelty of this study. The data analysis of monounsaturated fatty acids is not covered in this paper.

Point 2: Materials and methods:

Line 169: are the LC methods for the determination of monosaccharide composition worked out in house or they are adopted from another paper with modifications ? How was done the quantitative analysis? This needs to be described in the manuscript.

Response 2: Acid degradation combined with pre-column derivatization is the main method for polysaccharide determination of monosaccharide composition. And a large number of previous studies by our group have also shown the reliability of this method. Such as: 1.Cao YX, Li K, Qin XM, et al. Quality evaluation of different areas of Astragali Radix based on carbohydrate specific chromatograms and immune cell activities, Acta Pharmaceutica Sinica 2019, 54(7): 1277−1287.DOI: 10.16438/j.0513-4870.2019-0140.2.

  1. Shi LX, Li K, Qin XM, et al . Variety identification of different Bupleurum chinense based on sugar profile [J]. Journal of Pharmacy, 2020,55 (12): 2968-2975. DOI: 10.16438/J.0513.

In this paper, the absolute quantitative analysis of polysaccharides is not carried out, but only the relative content ratio of each monosaccharide composition is analyzed. Taking the peak area of glucose as the unit of 1.00, the relative content of other types of polysaccharides is calculated in turn. Thank you for your comments, this part has been supplemented in the article, see line 330.

Point 3:  Results and Discussion:

Line 400-402: Figure 4 – the legend is not clear. What mean M, -1L, -1M, -1H etc.?

The strengths and the limitations of the study should be specified.

Response 3: Thank you for your comments, the article has been revised to address this issue, see lines 218-221. M is the model group, PSDSs-1 low dose group (PSDSs-1L), PSDSs-1 middle dose group (PSDSs-1M), PSDSs-1 high dose group (PSDSs-1H).

Thank you for your suggestion, this paper has added the strengths and limitations of this study in the conclusion, see lines 599-615

Point 4: Technical issues:

  1. The whole text needs thoroughly editing.

Response 4: Thank you for your suggestion, this article has invited professional English staff to modify the content format of the article.